# An optimized GAN method based on the Que-Attn and contrastive learning for underwater image enhancement

**Zeru Lan, Bin Zhou** *, **Weiwei Zhao, Shaoqing Wang**

School of Computer Science and Technology, Shandong University of Technology, Zibo, Shandong, China

* freetzb@163.com

## Abstract

Research on underwater image processing has increased significantly in the past decade due to the precious resources that exist underwater. However, it is still a challenging problem to restore degraded underwater images. Existing prior-based methods show limited performance in many cases due to their reliance on hand-crafted features. Therefore, in this paper, we propose an effective unsupervised generative adversarial network(GAN) for underwater image restoration. Specifically, we embed the idea of contrastive learning into the model. The method encourages two elements (corresponding patches) to map the similar points in the learned feature space relative to other elements (other patches) in the data set, and maximizes the mutual information between input and output through PatchNCE loss. We design a query attention (Que-Attn) module, which compares feature distances in the source domain, and gives an attention matrix and probability distribution for each row. We then select queries based on their importance measure calculated from the distribution. We also verify its generalization performance on several benchmark datasets. Experiments and comparison with the state-of-the-art methods show that our model outperforms others.

**Data Availability Statement:** All data are available from the opening database. Data Availability Statement: Article data comes from public data sets. UIEB is available from URL: https://li-chongyi.github.io/proj_benchmark.html DOI:10.1109/TIP.

## Introduction

Compared with the depleted land energy, the use of the marine energy is still at its early stage. The rich marine energy contains not only abundant natural mineral energy such as petroleum but also sustainable renewable energy such as tidal and wave energy, which provides the possibility for the long-term development of human beings. In order to make full use of the marine energy, underwater image processing technology is critical. However, in the underwater environment, the captured images are often degraded by blurness, color deviation and low contrast. For example, as light travels through water, red light with longer wavelengths than green and blue light is absorbed more quickly, so underwater images often appear in a typical blue or green hue, As shown in Fig 1. In addition, large amounts of suspended particles tend to change the direction of light in the water, resulting in blurred images. Excellent underwater image enhancement is expected to improve the visual quality of input images by improving visibility, eliminating chromatic aberration and correcting low contrast. At the same time, enhanced

2019.2955241. HICRD is available from URL: https://github.com/JunlinHan/CWR DOI:10.3390/rs14174297. EUVP is available from URL: https://irvlab.cs.umn.edu/resources/euvp-dataset DOI:10.3390/rs14174297.

**Funding:** This work was supported by the Natural Science Foundation of Shandong Province under Grant ZR2021MF031 and ZR2020MF147. The funders had no role in study design, data collection and analysis, decision to publish, or preparation of the manuscript.

**Competing interests:** The authors have declared that no competing interests exist.

visibility can make scenes and objects of interest stand out, providing better preliminary information for high-level computer vision tasks such as object detection and recognition.

In the early stage of underwater image enhancement, pixels are reassigned in a single image. However, the redistributed image is prone to oversaturation and is highly dependent on prior knowledge. The emergence of deep learning solves these problems to a certain extent. However, underwater image algorithms based on deep learning usually require a large amount of datasets, especially paired datasets, but they are difficult to be collected. Studies have shown that underwater image enhancement using unsupervised learning can perform well in processing unpaired and unlabeled data sets. Of which, the contrastive method can compare the data with positive and negative samples in the feature space to obtain better sample feature representation. The deep learning model of generative adversarial network has a very powerful image enhancement effect. The combination of unsupervised contrastive loss and generative adversarial network can approximate the underwater enhanced image more closer to the real image. Therefore, this paper proposes to combine unsupervised contrastive learning with generative adversarial network for underwater image enhancement. Besides, an attention mechanism is introduced to further improve the underwater image enhancement effect. The main contributions of this paper are as follows:

- We propose an underwater image enhancement method based on unsupervised contrastive learning. This method combines unsupervised contrastive learning with generative adversarial network, which can maximize the mutual information between the original image and the restored image, loose the constraint on the large number of paired data sets, and produce clearer restored image.

- We propose an attention-querying mechanism for underwater image enhancement tasks. We select relevant anchor points first, and then use them as queries to focus on in order to absorb the features of other locations, and finally form better features suitable for

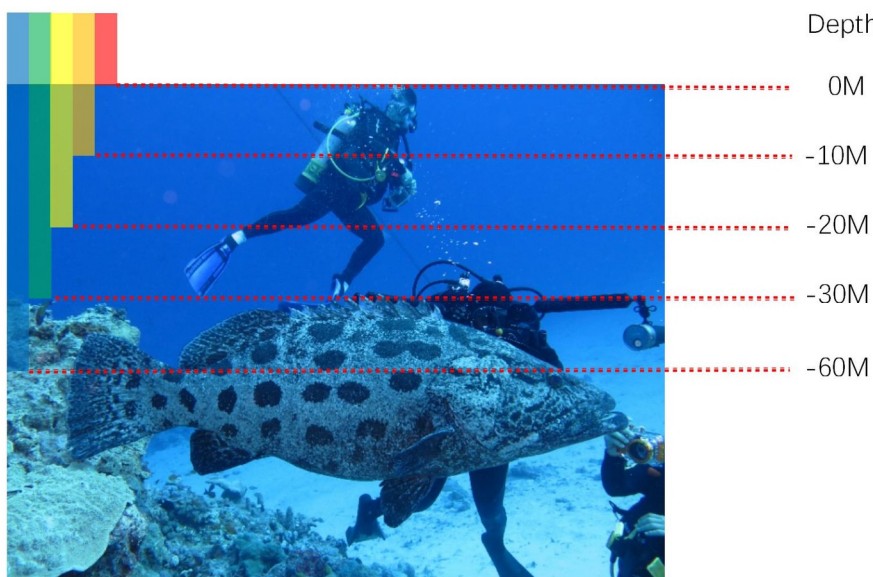

**Fig 1. Graphical demonstration of attenuation rates corresponding to different wavelengths when light propagates in water.** Blue travels the longest because it has the shortest wavelength. This is one of the main reasons why underwater images often appear blue [1].

comparative learning. Our proposed attention-querying mechanism preserves the simple design of CUT and does not increase any model parameters.

- We conduct multiple experiments on multiple public underwater data sets to prove our method can restore clearer images with higher quality of detail, and does not consume more computing resources. In summary, we prove that our method has competitive performance in terms of multiple evaluation indicators.

## Related work

In this part, we briefly review the existing works on underwater image enhancement and self-supervised learning. In the past few years, a variety of image enhancement methods have been proposed, which can be roughly divided into three categories: image processing-based methods [2–9], physics-based methods [1, 10–16] and learning-based methods [17–22].

### Underwater image enhancement

Image processing-based methods mainly adjust the pixel value of underwater images to improve the visual quality, including pixel value adjustment, retinal decomposition and image fusion. Zhang *et al*. [8] proposed an extended multi-scale Retinex underwater image enhancement method, including three steps, i.e. color correction, layer decomposition and enhancement for the input image. Ancuti *et al*. [9] proposed a new multi-scale fusion strategy that blends color compensation and white balance versions of a given image to produce better results. Recently, based on the severely uneven spectral distribution of underwater images, Ancuti *et al*. [10] proposed a new method which introduced a new color channel compensation preprocessing step in the opposite end color channel to overcome the artifacts. In summary, the image processing-based approach can improve the visual effect to a certain extent. However, they often fail to provide high-quality results in some complex scenarios due to ignorance of the domain knowledge in the field of underwater imaging.

Most physics-based approaches are based on underwater imaging models in which background light and transmission images are estimated by some prior. Prior knowledge includes underwater dark channel prior knowledge, minimum information prior knowledge and color line prior knowledge etc. Cosman *et al*. [13] proposed an underwater image restoration method combining fuzzy priori to estimate the scene depth more accurately. Inspired by the principle of minimum information loss, Li *et al*. [12] proposed a minimum information loss criterion. Guo *et al*. [12] estimated the optimal transmission graph to recover the underwater image and used histogram distribution to effectively improve contrast and brightness. Recently, Berman *et al*. [14] fused color line prior information and multispectral profile information of different water types into physical models, and optimized them using the grey world hypothesis theory, achieving good image denoising effect. These methods can restore underwater images well in some cases. However, when the prior information is invalid, some regions will inevitably suffer from unwanted artifacts and color deflections.

In recent years, with the development of deep learning, learning-based methods have made great progress in underwater image enhancement. There are many ways to improve performance by training their models on real underwater images. For example, in order to relax the need for paired training data, Li *et al*. [19] proposed a weakly supervised underwater color transfer model based on cyclic Uniform Generative Adversarial Network (CycleGAN) and real data. As a pioneering work, Li *et al*. [20] constructed a real underwater image enhancement dataset with a total of 950 pairs of original and reference underwater images. The reference

images were generated by 12 enhancement algorithms and scored by 50 volunteers to select the final result. With these images, Chen et al. [21] designed a threshold fusion network in which they learned three confidence graphs and fused the three preprocessed versions into an improved model. Recently, Li et al. [22] developed an underwater image enhancement network in a medium transport-guided multicolor space to achieve more robust enhancement. In a word, the restoration effect of the above methods on color and texture details is still unsatisfactory, and may produce unrealistic results. Luckily, we have achieved positive results by combining contrastive learning with attention mechanism.

There are also many algorithms that use data generated from generative adversarial networks or physical models to train their networks. For example, combining with the domain knowledge of underwater imaging, Li et al. [17] designed a generative adversarial network for generating underwater images from aerial images and depth maps, and then using these generated data to correct color projections in a supervised manner. Fabbri et al. [18] directly used CycleGAN to generate pairs of training data and then trained the full convolution codec to improve underwater image quality. In addition, Li et al. [19] proposed to synthesize 10 types of underwater images based on the underwater imaging model and some scene parameters. Using those synthetic data, Li et al. [20] proposed a new approach. Anwar et al. [22] proposed an end-to-end model that first directly restores a clear underwater dive image and then performs post-processing to improve subjective visual effect. Dudhane et al. [23] improved their work by introducing target fuzziness and color shift components to synthesize more accurate underwater image data.

## Self-supervised learning

Despite the great success of supervised learning, deep neural networks have been criticized for requiring large amounts of labeled training data. Recent research on self-supervised learning has shown that it has a strong ability to represent unlabeled images, especially with the help of contrastive loss [24, 25]. The idea is to extract features from the same image and push out features from different images to perform instance level discrimination and learn feature embedding. More recently, it has been studied as a pre-training technique that provides an initial model or potential embedding for underlying computer vision tasks [26–31]. Self-supervised learning is also applied to image generation. SS-GAN [32] uses rotation prediction as an auxiliary task of the discriminator to prevent over-fitting that is due to the limited true/false binary classification data. LT-GAN [33] trains an auxiliary classifier based on the embedded discriminator to classify whether the two pairs of false images have the same disturbance in the sampling noise vector. In addition to CUT, Kang et al. also adopted I2I's self-comparison learning. It uses a non-local [34] attention matrix to distort the target image to the source attitude and requires the features in the distorted image to approach the source through contrastive loss.

## Problem setup

Underwater small target detection and recognition based on optics is the key to the intelligent operation of the underwater fishing robot. However, underwater target detection and recognition technology based on optical vision also face significant challenges. The main reason is that the underwater image obtained by the visual vision system is seriously degraded due to the complex ocean imaging environment (the decline of underwater images mainly includes color deviation caused by underwater absorption of light-wave, refraction of light caused by forwarding scattering, and backward scattering, blurring of the imaged image, low contrast, obstruction of light by particulate matter in water, etc.), there are phenomena such as color fading, low contrast, and blurred details.

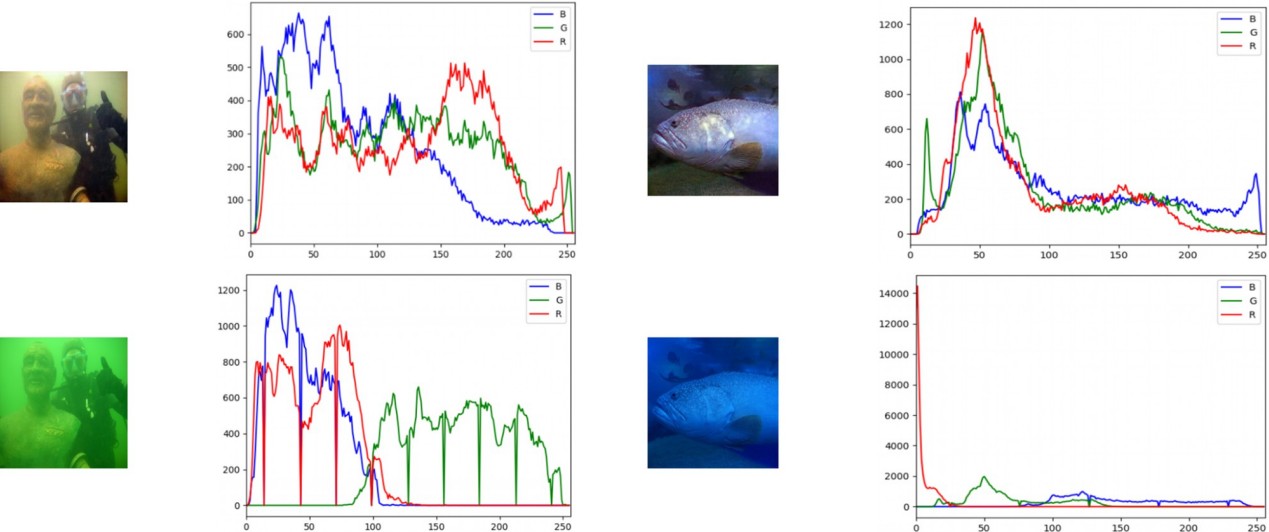

**Fig 2. Comparison of RGB tricolor histograms for underwater image and underwater restoration image.** Of which, the lower left figure is the input image, the upper left figure is the recovered underwater image, and the bottom right image is the input underwater image, the top right image is the recovered underwater image.

The underwater imaging equipment tells the development of underwater imaging, but it is more necessary to acquire high-quality underwater images using the state-of-the-art equipment in underwater exploration. The general imaging system will have the phenomenon of color degradation, loss of image detail texture, and blur in image imaging. More importantly, high-quality image acquisition requires a lot of costs. Underwater image enhancement is significant for obtaining high-quality underwater images for research. The Fig 2 shows the color change of the underwater image with the increase in seawater depth.

## Materials and methods

We define two fields $X \in \mathbb{R}^{H \times W \times C}$ and $Y \in \mathbb{R}^{H \times W \times C}$, given an image $I_x \in \mathbb{R}^{H \times W \times C}$ from source domain $X$ which represents real images, given an image $I_y \in \mathbb{R}^{H \times W \times C}$ from recovery domain $Y$. And our goal is to find the mapping $G : X \to Y$ to achieve underwater image recovery. Our model consists of a generator $G$ and a discriminator $D$, of which $G$ realizes the mapping from domain $X$ to domain $Y$, while $D$ guarantees that the translated image belongs to the right image domain. We denote the first part of the generator as an encoder and the second part as a decoder, written as $G_{enc}$ and $G_{dec}$. In the mapping process, we extract the features of the image using several layers of the encoder before passing the extracted features to a 2-layer *MLP* projection head (represented by function H). Here the projection head learning projects features extracted from the encoder onto a bunch of features. As shown in Fig 3, we give the schematic diagram of our model.

### Attention mechanism

For neural network learning, it is known to all, that the more parameters the network has, the stronger the representative ability the network exhibits, the more information the network stores, although this will incur information overload problem. Therefore, the attention mechanism can be introduced to focus on the feature which is more important to the task among numerous input features. By eliminating the need to pay attention to unimportant features, or

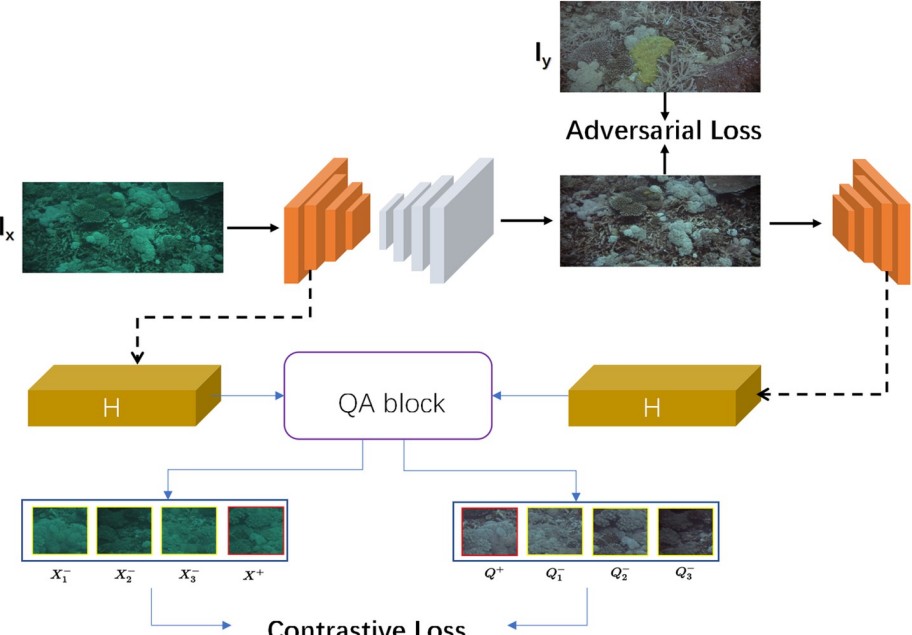

**Fig 3. The overall structure of our model.** The first step is to input the underwater image of the degradation domain, reconstruct the features through the generator $G(I_x)$, and extract the features from the two images by the encoder $G_{enc}$. Then the Que-Attn module selects the important features to establish the contrastive loss. The generated image is input to the discriminator for identification, and then the parameters of the whole network are updated.

filtering out features that have no contribution to the ground truth, we can therefore achieve data dimension reduction, and further promote the completing time and accuracy of the current task.

**Querying attention.** In the process of image processing, we introduce the procedure of attention querying for our underwater image enhancement model on the basis of unsupervised contrastive learning. Feature $F_x$ and $F_y$ is extracted from $I_x$ and $G(I_x)$ of encoder $G_{enc}$. And then we reconstruct and calculate $F_x$ to obtain the attention matrix $A_g$. In $A_g$, each row is sorted by significance level, and $N$ rows are selected to form query attention matrix $A_{QA}$. We then further apply routing to the feature value of the source and target domain, and obtain positive, negative and anchor features to construct the contrastive loss $L_{con}$. The positive and negative features are from the real image $I_x$, and the anchor features are from the translation image $G(I_x)$. Orange, blue, and green patches represent positive, negative, and anchor points, respectively. Some features do not reflect domain characteristics and are often retained during the transformation process. Therefore, the $L_{con}$ applied to them is not important to $q$. Our goal is to select the anchor point $q$ and compute $L_{con}$ at important anchor points that contain more domain-specific information. Our other goal is to define a quantified value for each potential location that reflects the significance of the feature. The quadratic attention matrix is used because it accurately reflects the similarity of each feature to others since it makes exhaustive comparisons with all other positions. As shown in Fig 4, we give the schematic diagram of attention block.

**Quering sets filtering.** The CUT network randomly selects anchor, positive sample, and negative sample to calculate the contrastive loss, which is inefficient because it needs to calculate the distance between positive samples and all negative samples, and their corresponding patches may not come from domain-related regions. Note that some features do not reflect

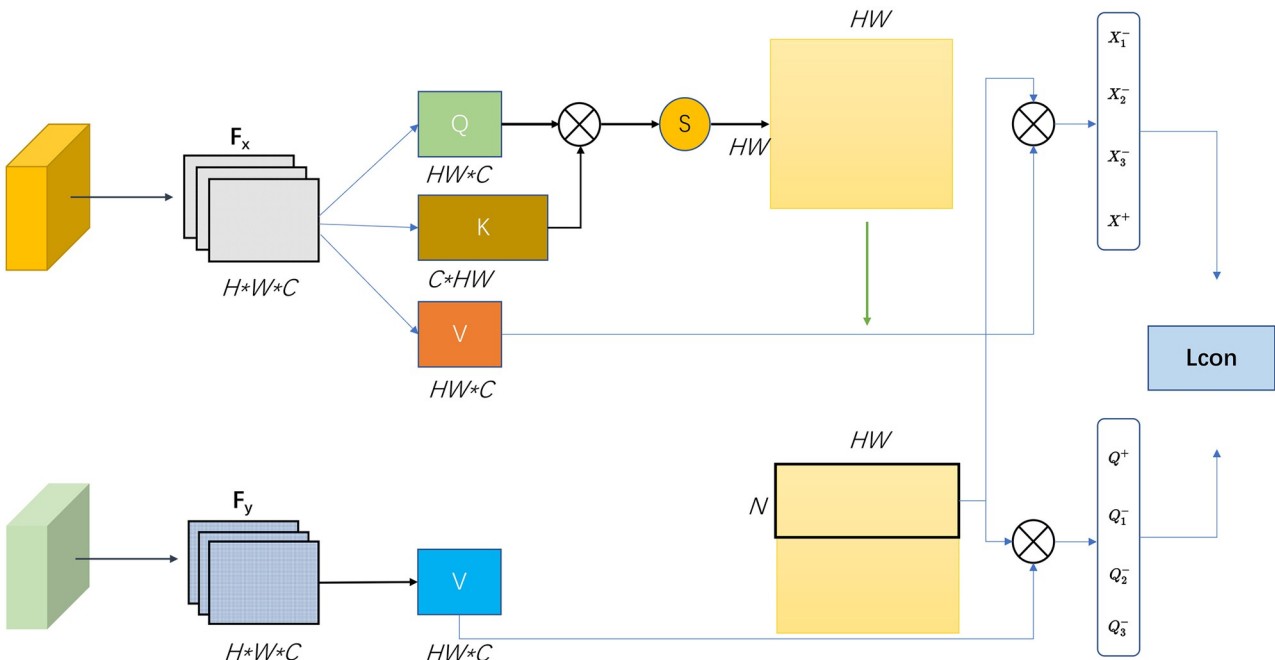

**Fig 4. Detailed diagram of attention block.** We first use the encoder to extract the features $F_x$ and $F_y$ of the input picture and the generated image. Then the attention matrix $A_g$ is obtained by matrix operation on $F_x$. We recombine the features according to their importance and select n rows to form the attention matrix $A_{QA}$ we need. The $A_{QA}$ is used to choose the patch between the source domain and the target domain value features. The positive, negative, and anchor features are obtained to construct the contrastive loss $L_{con}$. Both positive and negative samples are from the real domain image $I_x$, while the anchor is from the translated image $G(I_x)$.

domain characteristics and are often retained during the transformation process. Therefore, the contrastive loss imposed on them is not important. Our goal is to select anchor and calculate contrastive loss at important anchors that contain more domain-specific information.

**Global attention and local attention.** Based on the above observations, our goal is to define a quantified value for each potential location that reflects the importance of that feature. The quadratic attention matrix is used because it exhaustively compares each feature with all the others, and can accurately reflects its similarity with the other features. In particular, given a feature $F_x \in \mathbb{R}^{H \times W \times C}$ in the source domain, we first reshape it into a 2D matrix $Q \in \mathbb{R}^{HW \times C}$, and then multiply it by its transpose $K \in \mathbb{R}^{C \times HW}$. We then assign each row of the matrix a Softmax function to obtain the global attention matrix $A_g \in \mathbb{R}^{HW \times HW}$. Therefore, important characteristics can be measured according to the entropy of each row in the global attention matrix, which is computed as in Eq 1.

$$M_g(i) = -\sum_{j=1}^{HW} A_g(i,j) \log A_g(i,j) \tag{1}$$

Of which $i$ and $j$ are the indexes of the query and key, corresponding to the row number and column number in $A_g$. $M_g(i)$ approaches 0 means that there are very few key locations in row that are similar to query. Therefore, we assume that it is sufficiently significant and, importantly, constrained by the contrastive loss. In order to select all meaningful queries, the $A_g$ rows are sorted in ascending order by entropy $M_g$, and the smallest $N$ rows are selected as the query attention matrix $A_{QA} \in \mathbb{R}^{N \times HW}$. Note that $A_{QA}$ is fully determined by features in $I_x$, and is not relevant to $G(I_x)$.

Although global attention can help to obtain the global context and smooth the detailed context around the query, the computation cost of the query is high. Therefore, we combine the global attention and local attention in order to reduce computational cost. Local attention measures the similarity between the query and its adjacent keywords in a fixed window of $w \times w$ and step size of 1, which can capture the spatial interaction of local areas and reduce the computation cost. Given the reconstructed query matrix $Q_l \in \mathbb{R}^{HW \times C}$, we multiply it by the local keyword matrix $K_l \in \mathbb{R}^{HW \times w^2 \times C}$ and send it to the Softmax function to obtain the local attention matrix $A_l \in \mathbb{R}^{HW \times w^2}$. Local entropy $M_l$ is calculated in each row, as shown in the following Eq 2.

$$M_l(i) = -\sum_{j=1}^{w^2} A_l(i,j) \log A_l(i,j) \tag{2}$$

Here $i$ and $j$ are the indexes of the query and key. We select the smallest $N$ rows in $A_l$ by sorting $M_l$ in ascending order to form query attention matrix $A_{QA}$. For value routing, we also locate $N$ indices in the local value matrix $V_l \in \mathbb{R}^{HW \times w^2 \times C}$ and obtain the value matrix $V_{ls} \in \mathbb{R}^{N \times w^2 \times C}$.

## Loss function

The adversarial loss is used to encourage the generator to generate images that are visually similar to the target domain image. For the mapping $G : X \rightarrow Y$ with discriminator $D$, the loss of the generative adversarial network is calculated as follows:

$$\mathcal{L}_{GAN} = E_{x \sim p_{\text{data}}(x)}[\log D(x)] + E_{z \sim p_{\text{noise}}(z)}[\log(1 - D(G(z)))] \tag{3}$$

where $G$ tries to obtain the same image $G(x)$ as the image in the $Y$ domain, while $D$ aims to distinguish the generated sample $G(x)$ from the real data $Y$. Our goal is to maximize the mutual information between the patches corresponding to the input and output. For example, for the patch in the generated recovery image, we should be able to associate it more strongly with the same patch in the original input underwater image than with other patches in the image. Therefore, we use a noisy contrastive estimation framework to maximize the mutual information between the inputs and outputs. The basic idea behind contrastive learning is to connect the two kinds of information, and the contrastive method is to learn the characteristic representation of samples by comparing the data with positive samples and negative samples in the feature space.

We map the query, positive number and N negative numbers to a k-dimensional vector, which are respectively represented as $v$, $v^+ \in R^K$, and $v^- \in R^{N \times K}$. Note that $v_n^- \in R^K$ is the $n^{th}$ negative number. We establish a $(N + 1)$ classification problem and calculate the probability that a "positive" is chosen over "negatives". Mathematically, this can be expressed as a cross entropy loss, calculated by follow Eq 4.

$$l(v, v^+, v^-) = -\log \left( \frac{\exp\left(\frac{\text{sim}(v, v^+)}{\tau}\right)}{\exp\left(\frac{\text{sim}(v, v^+)}{\tau}\right) + \sum_{n=1}^{N} \exp\left(\frac{\text{sim}(v, v_n^-)}{\tau}\right)} \right) \tag{4}$$

Of which $\text{sim}(u, v) = v^T v / \|u\| \|v\|$ is the cosine similarity between $u$ and $v$. $\tau$ denotes the temperature parameter used to scale the distance between queries and other instances. We use 0.07 as the default value of $\tau$ and 255 as the default negative number.

We use a 2-layer projection head H to extract features from domain $X$. We first select $L$ layers from $G_{GEN}(X)$ to send to H, embedding an image into the feature stack $\{z_l\}_L = \{H^l(G^l_{enc}(x))\}_L$. It represents the output of the selected $L$ layers. After obtaining a bunch of features, each feature actually represents a patch in the image. Therefore, we denote the spatial position in each selected layer as $s \in \{1, \ldots, S_l\}$, where $l$ is the number of spatial positions of each layer. We select one query at a time and denote the corresponding positive feature as $Z^S_l \in R^{c_1}$, and all other negative features as $Z^{S/s}_l \in R^{(sl-1)\times c_1}$, where $c_1$ is the number of channels in each layer. Our goal is to match the patches corresponding to the input and output images. Therefore, we can use the following equation to represent the block-based multilayer contrastive loss of the mapping $G : X \rightarrow Y$.

$$\mathcal{L}_{PatchNCE}(G, H, X) = E_{x \sim X} \sum_{l=1}^{L} \sum_{s=1}^{S} l(\hat{z}^s_l, z^s_l, z^{S/s}_l) \tag{5}$$

The generated restored image should be realistic $L_{GAN}$, and the patches in the corresponding input original image and generated restored image should share a corresponding $L_{PatchNCE}$. The resulting restored image should share the same structure as the original input image, and the overall loss is:

$$\mathcal{L}(G, D, H) = \lambda_{GAN}\mathcal{L}_{GAN}(G, D, X) + \lambda_{NCE}\mathcal{L}_{PatchNCE}(G, H, X) \tag{6}$$

## Experimental setup

### Datasets

In our experimental setup, we conduct experiments on multiple datasets to test our model's performance and verify the robustness of our model on different datasets.

**UIEB** The UIEB dataset [20] is a composite dataset that consists of 890 paired images, with another 60 challenging images to be restored without ground-truth values. Among them, we select a random subset of 800 images as the training set, and the remaining 90 images as the test set. The sizes of the images are $860 \times 590$.

**HICRD** HICRD [35] is a composite dataset that contains 9, 676 original underwater images and 2, 000 restored reference images. It contains two subsets, unpaired HICRD and paired HICRD. It also contains measured parameters such as diffuse attenuation coefficients and camera sensor responses.

**EUVP** [36] This dataset is a composite dataset that is collected by various cameras, such as GoPros, micro-light USB, and other cameras during ocean exploration under varying visibility conditions. It includes 5, 550 paired test data, 3, 200 pairs of unpaired test data, and 515 pairs of test datasets. The resolution of the images is $256 \times 256$. We use 3, 200 unpaired data to train our network.

### The evaluation metrics

We consider two commonly used computer vision evaluation indicators, peak signal-to-noise ratio (PSNR) and structural similarity index measure (SSIM), to quantitatively analyze the performance of our model. PSNR calculates the quality of the generated image by the sum of the mean squared errors (MSE).

In order to verify the performance of underwater color recovery, we also use the no-reference underwater image quality measure (UIQM) [37] to analyze the quality of the output image. UIQM includes three underwater image property measures: underwater image

**Table 1. Quantitative results on synthetic underwater images in terms of SSIM, PSNR, and UIQM for EUVP dataset.**

| Method | PSNR | SSIM | UIQM |
|---|---|---|---|
| WaterNet [20] | 24.43 ± 4.64 | 0.82 ± 0.08 | 2.97 ± 0.32 |
| FUnIE-GAN [36] | 26.19 ± 2.87 | 0.82 ± 0.08 | 2.84 ± 0.46 |
| DeepSESR [38] | 25.30 ± 2.63 | 0.81 ± 0.07 | 2.95 ± 0.32 |
| Shallow-UWnet [39] | 23.52 ± 2.70 | 0.83 ± 0.07 | 2.98 ± 0.38 |
| CycleGAN [40] | 24.30 ± 2.53 | 0.93 ± 0.01 | 3.02 ± 0.08 |
| **Ours** | **26.55 ± 2.27** | **0.95 ± 0.01** | **3.14 ± 0.01** |

colorfulness measure (UICM), underwater image sharpness measure (UISM), and underwater image contrast measure (UIConM). Each attribute evaluates one aspect of underwater image degradation.

## Experimental parameter setting

Our network is trained for 200 epochs at the same learning rate of 0.0002. The speed of learning declines linearly after half an epoch. We load all images with a resolution of 800 × 800 and divide them randomly into 512 × 512 blocks during training. We use images of 1680 × 892 resolution for all methods. For our network, we use spectral normalization for discriminators and instance normalization for generators. We set the batch size to 1, and use ADAM optimizer for optimization. We set $\beta_1 = 0.5$ and $\beta_1 = 0.999$. We use the $Tesla V100 - 32GB$ GPU to train our method and other baselines.

## Results

In order to objectively evaluate the restoration quality of our method, SSIM and PSNR are selected to evaluate the enhancement performance on EUVP datasets partially-restored images, degradated images and clear images. At the same time, the underwater image quality evaluation index UIQM is introduced to measure the color measure (UICM), sharpness measure (UISM) and contrast measure (UIConM) of the generated image. The results are shown in Table 1, and we can see that our method has obtained good results.

At the same time, we conduct experiments on UIEB datasets, among which 3200 pairs of unpaired datasets are selected for training and 515 test datasets are used for testing. The obtained results exhibit good performance in SSIM, PSNR and UIQM, as shown in Table 2.

In order to verify the generalization ability of our model, we also conduct experiments on the high-resolution dataset HICRD. The HICRD dataset has a resolution of 1842 × 980. We analyze our experimental results quantitatively and qualitatively, and show that our method

**Table 2. Quantitative results on synthetic underwater images in terms of SSIM, PSNR, and UIQM for UIEB dataset.**

| Method | PSNR | SSIM | UIQM |
|---|---|---|---|
| WaterNet [20] | 23.11 | 0.79 | 3.02 |
| FUnIE-GAN [36] | 20.13 | 0.79 | 2.99 |
| Deep SESR [38] | 19.26 | 0.73 | 2.95 |
| Shallow-UWnet [39] | 18.99 | 0.81 | 2.77 |
| FUnIE-GAN-UP [36] | 25.22 | 0.78 | 2.93 |
| **Ours** | **25.65** | **0.89** | **3.04** |

**Table 3. Quantitative results on synthetic underwater images in terms of SSIM, PSNR, and UIQM for HICRD dataset.**

| Method | PSNR | SSIM | UIQM |
|---|---|---|---|
| DCP [41] | 14.27 | 0.532 | 2.22 |
| Haze-line [14] | 14.69 | 0.423 | 2.45 |
| UDCP [42] | 13.31 | 0.493 | 3.22 |
| IBLA [13] | 19.42 | 0.463 | 2.42 |
| CUT [43] | 26.30 | 0.796 | 5.26 |
| CycleGAN [40] | 21.82 | 0.591 | 5.27 |
| UWGAN [18] | 26.55 | 0.811 | 4.89 |
| DCLGAN [44] | 21.92 | 0.732 | 5.11 |
| CWR [35] | 26.88 | 0.831 | 5.43 |
| **Ours** | **26.89** | **0.897** | **5.56** |

not only lead in many evaluation metrics, but also exhibit good visual performance as shown in Table 3. And we give the visual effect as shown in Fig 5.

## Ablation experiments

In order to further prove the superiority of our network structure, we conduct experiments on different modules of our proposed network. We mainly focus on: 1 whether contrastive

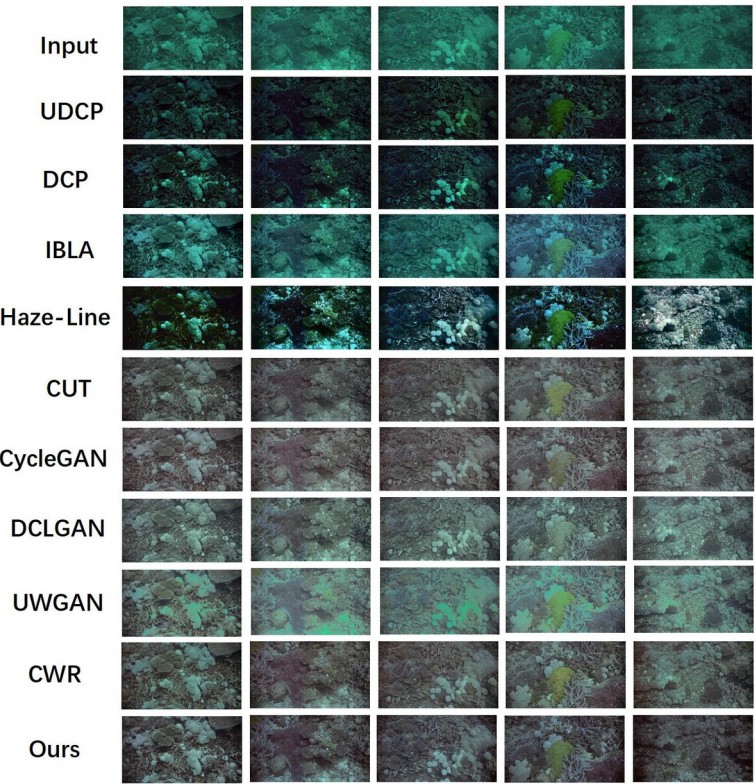

**Fig 5. Qualitative results on the HICRD test dataset, in which all examples are randomly selected from the test dataset.** We compare our model with other underwater image restoration baselines. Traditional restoration methods cannot remove underwater images' green and blue tones. Our model shows satisfactory visual effects without content loss and structure loss.

**Table 4. Quantitative results on synthetic underwater images in terms of SSIM, PSNR, and UIQM.** A represents the use of ordinary GAN network, B represents the use of GAN network and contrastive loss, and C represents the use of GAN network, query attention and contrastive loss.

| Method | PSNR | SSIM | UIQM |
|--------|------|------|------|
| A | 23.11 | 0.79 | 3.02 |
| B | 20.13 | 0.79 | 2.99 |
| C | **25.65** | **0.89** | **3.04** |

learning is used and 2 the use of query attention mechanism. We test the effect of these modules on the experimental results. Table 4 shows the experimental results. And from it, we can see our model has asymptotic property.

## Conclusion

This paper proposes an underwater image enhancement model. The network is an end-to-end unsupervised generative adversarial network. This network uses contrastive learning and attention mechanism to complement the original network. Our qualitative and quantitative analysis of multiple datasets show that our network has competitive results in the generated recovered images. The experiments on real datasets also show that our network is more robust.

## Author Contributions

**Data curation:** Zeru Lan, Bin Zhou, Weiwei Zhao, Shaoqing Wang.

**Funding acquisition:** Bin Zhou.

**Methodology:** Zeru Lan, Bin Zhou, Weiwei Zhao, Shaoqing Wang.

**Validation:** Zeru Lan, Bin Zhou, Weiwei Zhao, Shaoqing Wang.

**Writing – original draft:** Zeru Lan, Bin Zhou, Weiwei Zhao, Shaoqing Wang.

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
