## [Decision Letter · Decision Letter 0]

24 Oct 2022

PONE-D-22-22875An Optimized GAN Method Based on the Que-Attn and Comparative Learning for Underwater Image EnhancementPLOS ONE

Dear Dr. Zhou,

Thank you for submitting your manuscript to PLOS ONE. After careful consideration, we feel that it has merit but does not fully meet PLOS ONE’s publication criteria as it currently stands. Therefore, we invite you to submit a revised version of the manuscript that addresses the points raised during the review process. Here are two comments, one is a major revision, the other is a minor revision. Please revise it carefully.

We look forward to receiving your revised manuscript.

Kind regards,

Yiming Tang, Ph.D.

Academic Editor

PLOS ONE

Journal Requirements:

"YES - Specify the role(s) played."

“This work was supported by the Natural Science Foundation of Shandong Province under Grant No. ZR2021MF031 and No. ZR2020MF147”

“YES - Specify the role(s) played.”

6. We note that Figure 2 includes an image of a participant in the study.

Reviewers' comments:

Reviewer's Responses to Questions

**Comments to the Author**

1. Is the manuscript technically sound, and do the data support the conclusions?

Reviewer #1: Yes

Reviewer #2: Yes

2. Has the statistical analysis been performed appropriately and rigorously? 

Reviewer #1: Yes

Reviewer #2: No

3. Have the authors made all data underlying the findings in their manuscript fully available?

Reviewer #1: Yes

Reviewer #2: Yes

4. Is the manuscript presented in an intelligible fashion and written in standard English?

Reviewer #1: No

Reviewer #2: Yes

5. Review Comments to the Author

Reviewer #1: The work in the paper looks solid with good results. Unfortunately the English could be improved.

- In a large number of places the paper talks about contrast learning rather than contrastive learning

- Line 172 refers to Fig 3-2. There is no such figure.

- Fig 3 has missing labels for G(I_X), I_X, I_Y, F_X, F_Y, A_g and A_{QA} which makes it hard to understand

- Fig 4 also missing labels, e.g F_X and F_Y

- In the caption of Fig 4 it uses "mg" presumably for "A_g"

- Line 183: "Our another" should be "Our other"

- Line 194: The sentence ending "not important to ." seems unfinished

- Line 250: "We use and a" should be "We use a"

Generally the paper is let down by the English which makes it quite hard to understand.

Reviewer #2: - Line 92 « Li et al [21] », Reference 21 is not authored by Li et al. Please correct.

- Lines 96-98: It is said that “training on real data can produce visually satisfactory results”, but also that “they often produce unreal results because the reference image is not the actual ground truth” How real data could not be actual ground truth ?

- Line 172 “The background theory is shown in Figure 3-2”.There is no Figure 3-2. Please correct

- Fig.2 is not documented in the core of the manuscript. We don’t know exactly what the purpose of this figure is. Please specify where the original underwater images and the restored images are.

- Qualitative results on the HICRD test dataset (fig. 5) and Quantitative results are given for EUVP dataset (table 1). It would be interesting to have the two evaluations for the same dataset to compare the qualitative and quantitative results.

- In figure 5, the authors claim the superiority of their method compared to traditional restoration methods. But the CUT, CycleGAN and CWR methods are visually sometimes more satisfying than the proposed method.

- Tables1, 2, 3, the experimental data,(EUVP, HICRD, UIEB) are presented as “synthetic underwater images”, whereas in Datasets paragraph (p.9/15), they are not.

- The problem of time efficiency is mentioned in page 10 (“speed and memory” paragraph) but real time processing is not tackled, it is pivotal for robots’ autonomous operations. Thus, it is essential to develop a real-time and adaptive method for underwater visual restoration. What is the real-time efficiency of the proposed method and how could it be improved.

- To provide complete information, the legends of the tables of results and the figures of results must specify the data used for learning and the test data.

- Table 4 should be more clearly described. What is the difference between A (Ordinary GAN Network) et C (GAN Network) ?

- The proposed attention-querying mechanism preserves the simple design of CUT and does not add any model parameters. However, in reference 41 (Park T, Efros AA, Zhang R, Zhu JY., European conference on computer vision. Springer; 2020. p. 319–345), the authors show results (see Fig. 3 in the cited article) with two failure cases. In the first case, they were unable to identify the unfamiliar pose of the horse and instead add texture to the background. In the second, the method added a tongue to the dog that does not exist. Have you been faced with such failure situations. If not, could you explain why.

6. PLOS authors have the option to publish the peer review history of their article (what does this mean?). If published, this will include your full peer review and any attached files.

Reviewer #1: No

Reviewer #2: **Yes: **Abdelkader Boulanouar

---

## [Author Response · Author response to Decision Letter 0]

28 Nov 2022

Dear Editor,

Thank you for allowing a resubmission of our manuscript, with an opportunity to address the reviewers’ comments.

We are uploading (a) our point-by-point response to the comments (Response to Reviewers) and (b) an updated manuscript with yellow highlighting indicating changes (Revised Manuscript with Track Changes) and (c) an updated manuscript without yellow highlighting indicating changes (Manuscript).

Best regards,

Bin Zhou et al.

---

## [Decision Letter · Decision Letter 1]

19 Dec 2022

An Optimized GAN Method Based on the Que-Attn and Contrastive Learning for Underwater Image Enhancement

PONE-D-22-22875R1

Dear Dr. Zhou,

We’re pleased to inform you that your manuscript has been judged scientifically suitable for publication and will be formally accepted for publication once it meets all outstanding technical requirements.

Kind regards,

Yiming Tang, Ph.D.

Academic Editor

PLOS ONE

Additional Editor Comments (optional):

Reviewers' comments:

Reviewer's Responses to Questions

**Comments to the Author**

1. If the authors have adequately addressed your comments raised in a previous round of review and you feel that this manuscript is now acceptable for publication, you may indicate that here to bypass the “Comments to the Author” section, enter your conflict of interest statement in the “Confidential to Editor” section, and submit your "Accept" recommendation.

Reviewer #1: All comments have been addressed

Reviewer #2: All comments have been addressed

2. Is the manuscript technically sound, and do the data support the conclusions?

Reviewer #1: Yes

Reviewer #2: (No Response)

3. Has the statistical analysis been performed appropriately and rigorously? 

Reviewer #1: Yes

Reviewer #2: (No Response)

4. Have the authors made all data underlying the findings in their manuscript fully available?

Reviewer #1: Yes

Reviewer #2: (No Response)

5. Is the manuscript presented in an intelligible fashion and written in standard English?

Reviewer #1: Yes

Reviewer #2: (No Response)

6. Review Comments to the Author

Reviewer #1: The authors have made substantial changes improving the paper. I am now happy that this is of sufficient quality to be accepted.

Reviewer #2: (No Response)

7. PLOS authors have the option to publish the peer review history of their article (what does this mean?). If published, this will include your full peer review and any attached files.

Reviewer #1: No

Reviewer #2: **Yes: **Abdel-Kader Boulanouar

---

## [Editor Report · Acceptance letter]

27 Dec 2022

PONE-D-22-22875R1 

An optimized GAN method based on the Que-Attn and contrastive learning for underwater image enhancement 

Dear Dr. Zhou:

I'm pleased to inform you that your manuscript has been deemed suitable for publication in PLOS ONE. Congratulations! Your manuscript is now with our production department. 

Kind regards, 

on behalf of

Professor Yiming Tang 

Academic Editor

PLOS ONE